# Development and Evaluation of the Chromatic Behavior of an Intelligent Packaging Material Based on Cellulose Acetate Incorporated with Polydiacetylene for an Efficient Packaging

**DOI:** 10.3390/bios10060059

**Published:** 2020-05-31

**Authors:** Lina D. Ardila-Diaz, Taíla V. de Oliveira, Nilda de F. F. Soares

**Affiliations:** 1Program of Agroindustrial Engineering, Faculty of Agricultural Engineering, Universidad del Tolima, Ibagué-Tolima 730006299, Colombia; 2Packaging Laboratory, Department of Food Technology, Universidade Federal de Viçosa, Viçosa-MG 36570-900, Brazil; taila.oliveira@ufv.br (T.V.d.O.); nilda.soares@ufv.br (N.d.F.F.S.)

**Keywords:** polydiacetylene, smart packaging, quality food, safety food, sensor

## Abstract

Global growth of the food industry and the demand for new products with natural characteristics, safe conditions and traceability have driven researches for the development of technologies such as intelligent packaging, capable to fulfil those needs. Polydiacetylene (PDA) is a synthetic material that has been highlighted in research field as a sensor substance, which can be used to produce intelligent packaging capable to detect chemical or biochemical changes in foods and in their environment due to PDA’s color transition from blue to red. This work focused on the development and optimization of an intelligent packaging constituted of a polymeric matrix of cellulose acetate-based incorporated with PDA as the substance sensor. Cellulose acetate films (3% wt.) were developed by a casting method, and the amounts of triethyl citrate plasticizer (TEC) (0–25% wt. of cellulose-acetate) and PDA (0–60 mg) were analyzed to optimize the conditions for the best color transitioning at this study range. The compound amounts incorporated into polymeric matrices were established according to Central Composite Designs (CCD). Three more design variables were analyzed, such as the polymerization time of PDA under UV light exposition (0–60 min), pH values (4–11) and temperature exposure on the film (0–100 °C), important factors on the behavior of PDA’s color changing. In this study, film thickness and film color coordinates were measured in order to study the homogeneity and the color transitioning of PDA films under different pH and temperature conditions, with the purpose of maximizing the color changes through the optimization of PDA and TEC concentrations into the cellulose acetate matrix and the polymerization degree trigged by UV light irradiation. The optimal film conditions were obtained by adding 50.48 g of PDA and 10% of TEC, polymerization time of 18 min under UV light, at 100 °C ± 2 °C of temperature exposure. The changes in pH alone did not statistically influence the color coordinates measured at the analyzed ratio; however, variations in pH associated with other factors had a significant effect on visual color changes, and observations were described. PDA films were optimized to maximize color change in order to obtain a cheap and simple technology to produce intelligent packaging capable to monitor food products along the distribution chain in real time, improving the food quality control and consumer safety.

## 1. Introduction

The food industry has focused on ensuring that product quality is in accordance to consumers’ requirements. Food packaging plays an essential role in finished products, influencing sales since the consumers’ choice of purchase is heavily based on the visual attribute of the packaging [1,2,3]. The customer will decide to buy or reject the product based on how attractive and informative the packaging is [4,5].

First and foremost, food packaging manufacturing is a function of the distribution and preservation of food products. The selection of the packaging materials depends on the properties of the packaged product and process conditions, including the steps of storage, distribution and consumption of food products. The packaging functionality is attributed to the capacity isolating the food from the external environment that speeds up the food degradation process, changing its original features [6]. The contact with the external environment can cause food contamination, due to the presence of microorganisms, fungi, viruses, physical and chemical materials, or due to process disturbances such as temperature, pH, light exposure and O_2_ presence, leading to product deterioration [3,7]. In addition to the regular functions of the packaging, it also serves as a communication instrument to inform about the product consumers and ingredients used to provide convenience [8,9].

One of these innovations is known as intelligent packaging, which is formed by a communication system capable to indicate the security and quality of the product in real-time. This device perceives the changes that have happened in the food or in its environment throughout the entire supply chain, ensuring the safety and reliability of food products in the market [10].

According to literature, the colorimeter sensors in the packaging have the potential to be used as a detector of product quality by tracking the presence of microorganisms [11,12], carbon dioxide [13], behavior changes such as temperature [14,15,16], pH [17], hermetic sealing, nutritional properties and degree of maturity [18,19]. A proper alert can avoid economic losses for the industry, and prevent any risks that may jeopardize consumer health [4,5,20].

Some colorimetric sensors can be produced using the polydiacetylene as a color change indicator compound. This material can be applied to monitor and to indirectly indicate metabolite presence that alter the food system or its internal atmosphere [21,22]. The polydiacetylene, also known as PDA, is a compound that has the potential to be applied into intelligent packaging as a colorimetric sensor. PDA has chromatic properties due to its capacity to change color from blue to red when exposed to external disturbances [23]. Intelligent packaging produced by PDA incorporation into a polymeric matrix is capable to change colors, similar to PDA as a free compound. PDA color behavior can be used to monitor the quality and safety of the food. Therefore, this study evaluated simultaneous factors that influence the capacity of PDA to change colors in a polymer matrix and the use of it as an intelligent food packaging produced with cellulose acetate. This technology needs standardization to precisely identify variations of food conditions and to ensure consumer health.

## 2. Materials and Methods

### 2.1. Materials

The 10,12-Pentacosadiynoic acid (PDA) (Merck^®^), acetone (Merck^®^), cellulose acetate (saturation degree = 2.5, MM = 2024.00 g.mol-1) from Rhodia Solvay Group (Santo André, SP, Brazil) and Trietil citrate (TEC) C12H20O7 from Merck^®^ (Viçosa, MG, Brazil) were used to produce the colorimetric sensor films. The monopotassium phosphate KH2PO4 (Merck^®^), sodium hydroxide NaOH (Merck^®^), acetic acid CH3COOH (Merck^®^), sodium acetate C2H3NaO2 (Labsynth^®^) and sodium bicarbonate NaHCO3 (Labsynth^®^) were used to prepare buffer solutions at different pH values. Milli-Q (>18 MΩ) Ultrapure water (Merk Millipore, Darmstadt, Germany) was used as the solvent liquid.

### 2.2. Manufacturing of the Polydiacetylene Sensor Film 

The intelligent films were produced using the casting method, according to Mohammad et al. [24] methodology. Cellulose acetate (2.5 g) was added in acetone (25 mL) and the solution was left to sit at room temperature (25 °C ± 2 °C) for 24 h. In this study, TEC plasticizer concentration varied from 0.00% to 25.00% wt., which was pre-determined according to the literature [25]; and PDA quantities varied from 0.00 mg to 60 mg, pre-established from pre-tests done, being added into the polymeric solution in concentrations defined by Central Composite Design (CCD), as described on item 2.5. The resulting film-forming solution was magnetically stirred (500 rpm) (LSPLabor^®^) for 5 min and suspended for 5 min at room temperature. The filmogenic solution was poured into (19 × 44) cm glass plates coupled to the K paint applicator (Printcoat instruments^®^). The films were dried for 10 h at room temperature (25 ± 2 °C) and subsequently removed and stored in multilayer films.

### 2.3. Optimization of PDA Sensor Films

The films were cut in (5 × 5) cm^2^ standard samples, and thickness values were obtained using a micrometer (Mitutoyo^®^, model 547-401, Japan). Ten random points of each treatment were measured, and the average results expressed in micrometers (μm). According to the CCD combination, the samples were irradiated with Ultraviolet light (UV) at 254 nm (constant flow, 15 W potency, 110 V, 08 × 07 × 25 cm of dimensions and 15 cm of distance work) during (1, 18, 30, 43 and 60) min, and the color measurements were done promptly after. Subsequently, they were immersed in 10 mL of buffer solution (final pH value of 1, 4, 7, 9 and 11), and stored at different temperatures (100, 71, 50, 28 and 0 °C) for 10 min, as established by the CCD combination. The temperature behavior was obtained using climate control with the climatic chamber, heater (Ethiktechnology^®^) and refrigerator (Consul^®^). Finally, their final color properties were immediately evaluated.

### 2.4. Color Analysis of PDA Sensor Films

Color analysis of the intelligent films were evaluated before and after exposure under different pH values and temperatures. In the evaluation of the color analysis, samples were fixed with adhesive tape support in glass transmission cells (50 × 55 × 57) mm. The colorimetric analysis was performed in the ColorQuest XE (HunterLab^®^) colorimeter, under conditions of illuminant D65 (average daylight) and 10° angle (field of view) according to the methodology of Norte et al. [26] and Silva et al. [27]. CIELAB scale was used and the coordinates evaluated were: *L** value (luminosity), which varies from 0 (Black) to 100 (White); the coordinates *a** and *b**, where *a** indicates the content of green (negative *a** values) to red (positive *a** values) and *b** registers the content of blue (negative *b** values) to yellow (positive *b** values) [28]. The color difference between the samples, before and after external exposure, were determined using the ΔE* equation (Equation (1)) and quantified through the terms Δ*L**, Δ*a** and Δ*b**. First, the samples exposed to ultraviolet light, which were considered the reference samples. The final values correspond to the samples exposed to the different pH values and temperatures [29].

Δ*E** means the color difference between samples before and after external exposure; *L** represents the brightness in the samples (from 0 to 100); ∆L* means the brightness difference between samples before and after external exposure; *a** means the coordinate (green: negative-values/red: positive-values) in the sample; ∆*a** means the difference of *a** coordinate in the sample; *b** means the coordinate (blue: negative-values/yellow: positive-values) in the sample; ∆*b** means the difference of the *b** coordinate/between samples before and after external exposure.
Δ*E* = [(*∆*L*)*^2^*+ (*∆*a*)*^2^*+(*∆*b*)*^2^*]*^1/2^(1)

### 2.5. Experimental Design

Central Composite Design (CCD) using Response Surface Methodology (RSM) in order to indicate the concomitant conditions for all of the factors on the colorimetric behavior analysis of PDA films. The five factors and two levels consisting of forty-five experimental runs were employed including three replicates in the center point. The effects of unexplained variability in the observed response due to experimental errors were minimized by randomizing the order of the experiments, while response variables were colorimetric coordinates *L**, *a**, *b** and the colorimetric difference determined by *ΔE**. The design variables were: temperature exposure, pH values, PDA irradiation-time with UV light, PDA concentration and TEC concentration. The effects of the factors on the smart films were evaluated through RSM using Minitab 17© software. Symbol codes, experimental range and levels of the independent variables were described in Table 1.

## 3. Results

Intelligent films were produced by the incorporation of polydiacetylene into cellulose acetate-matrix to detect changes in food or in food behavior through film color transitioning from blue to red. Casting method was used to produce intelligent films. Film thickness was analyzed to verify the homogeneity of the dry process, since heterogeneous samples could present problems on both mechanical and barrier properties. The thickness of the intelligent films was homogeneous, and the average was 0.026 ± 0.01 µm, regardless of PDA and TEC concentration that was added into the films (*p* > 0.05).

Statistical models were adjusted to describe the color coordinate L*, *a**, *b*,* L**_f_*, *a*_f_*, *b*_f_* and Δ*E** as a function of PDA and TEC concentration, UV irradiation time, pH changes and Incubation Temperature (Table 2). Films of cellulose-acetate that were incorporated with polydiacetylene, after UV irradiation and before exposure to different values of pH and temperature, assumed the blue color. The blue color occurred due to the polymerization of diacetylene compounds, causing the reduction of the L* and *b** coordinate values of PDA films through the enhancement of PDA and TEC concentrations (Figure 1a,c, respectively). The increment in the amounts of PDA improved the polymerization, since this compound is responsible for providing the color of intelligent films, and the increasing addition of TEC favored the plasticizer function, responsible for the diacetylene monomer mobility and polymerization [29]. The improvement in the movement of diacetylene monomers facilitated the rearrangement and the polymerization, forming the polydiacetylene compound, which has a planar form and a blue color [30]. The *a** coordinate was influenced by PDA concentration and UV light exposure after the minimum point; the values of *a** coordinate increased because of the colorimetric transition from blue to purple (Figure 6a,b). This behavior occurred because of the excess of energy provided by UV light, that was enough to cause a partial color change on the intelligent films.

The study followed with the exposure of the UV-irradiated-PDA films under different conditions of pH values and temperatures. Statistical models were adjusted as a function of the coordinate *b*_f_*, reporting a maximum point dependent on PDA and TEC amounts (Figure 2c). After the maximum point, the additions of TEC into the polymer matrix reduced the coordinate *b*_f_* because of the natural yellowish color of this compound. 

The *a*_f_* coordinate data reported a minimum point. After that, the increment of PDA and TEC amounts favored the color transition under different pH and temperature stimuli because of the enhancement in PDA’s mobility by TEC plasticizer, facilitating the chain change from planar to non-planar packaging; plus, more color was added into the matrix.

The color difference of PDA films, expressed by ΔE* factor, was mainly affected by the UV exposure-time, TEC addition into the intelligent films and temperature exposure. The UV exposure-time in excess, facilitated by the TEC plasticizer effect, caused a colorimetric transition from blue to purple, corroborating with *a** coordinate, since the excess of energy provided by UV light was enough to a partial color change of the intelligent films. UV-time optimization in the sensor performance is important, since the partial color transition occurred when this factor was long, meaning a false positive in the indication of food deterioration. 

Similarly, high temperature at a short UV-irradiation time induced a complete color change by thermochromism, enhancing ΔE* values (Figure 2 and Figure 3). High temperatures increased the vibration motions that provide enough energy to cause structural changes on PDA carbon chains [30,31,32,33]. Values of ΔE* higher than 3.5 can be visible to the naked eye, as previously achieved in this study at the condition of long UV irradiation time and high temperature, indicating that this intelligent packaging can be used successfully as a sensor of temperature conditions [34,35,36].

The color transitioning from blue to red can be observed in Figure 4 and Figure 5, even though pH and temperature values were not significant at 5% for the adjusted statistical models. The intelligent films that were immersed in a 4.0 pH solution, incorporated with 42.63 mg of PDA and 17.73% of TEC, and exposed during 18 min at UV light maintained blue color at 28.0 °C; however, they changed color at 71.1 °C, as shown in Figure 4.

Intelligent films incorporated with the highest PDA concentration hampered the color transition, since total polymerization impairs structural changes on polydiacetylene due to external stimuli such as temperature, UV or pH, and therefore, requires more energy to cause structural changes on the PDA chain (Figure 6).

Intelligent films at fixed conditions of temperature (71 °C) and PDA concentration (42.7 mg) after exposure to a pH-9.0 solution turned red, whereas an exposure to a pH-4.0 solution turned the films purple. This behavior corroborates with data in previous literature [32,37,38]. Although the color change caused by pH factor had not been statically perceivable, the difference can be noticeable to the naked eye, as it can be observed in Figure 5. The mechanism for the color transition accepted by the research community is the structural transition of PDA’s carbonic chain from planar to non-planar form when the compound is exposed to an external stimulus, and this transition is influenced by the lateral chain [39,40]. Chen et al. (2012) [41] observed that the color transitioning of vesicles produced only with tricosadinoic acid occurred after NaOH addition. The color changes occurred due to the deprotonation of the carboxylic acids, from the polymer polar groups, which increased Coulombic repulsive force, leading to a new zig-zag PDA’s polymeric carbonic chain [41,42,43]. Same behavior was observed for PDA films exposed to high pH conditions; however, steric impediment due to polymer matrix hampered the deprotonation of PDA carboxylic groups reducing the intensity of the color changes.

The color change of the intelligent films was optimized through the desirability function to maximize the color transition, establishing the highest values of ΔE*, *a***_f_* and *b*_f_.* The optimized conditions were achieved when the intelligent films were produced with 50.5 mg of PDA, 10% of TEC, after 18 min of UV-light exposure and at a temperature condition of 100 °C (Figure 7). These optimized conditions make it possible to use the sensor as a colorimetric indicator in real-time under high temperatures, such as in the pasteurization process of fruit juices, milk, milk derivates/dairy products, beer, ice cream and eggs [44,45,46,47].

This work contributed to the development and optimization of the intelligent packaging produced by the PDA incorporation into a polymer matrix, such as cellulose acetate, providing a simple and cheap technology to monitor the temperature of food products during the entire production chain.

## 4. Conclusions

The intelligent films produced by the incorporation of PDA into a cellulose-acetate matrix were capable to change the color of PDA films when exposed to the variation of pH solutions, high-temperature conditions and prolonged UV-light time. The concentration of PDA and TEC incorporation was optimized to maximize the color transitioning under the variation of external factors. The colorimetric variables were used to quantify the color change from blue to red and statistical models were adjusted to understand the influence of these factors in color behavior of PDA films. This development can be applied in intelligent packaging to detect high temperatures during the production chain, and these color changes can be highlighted by high pH values. Therefore, extreme conditions that cause food contamination and food quality loss can be detected by an intelligent packaging through the PDA incorporation into a biodegradable-polymer matrix.

## Figures and Tables

**Figure 1 biosensors-10-00059-f001:**
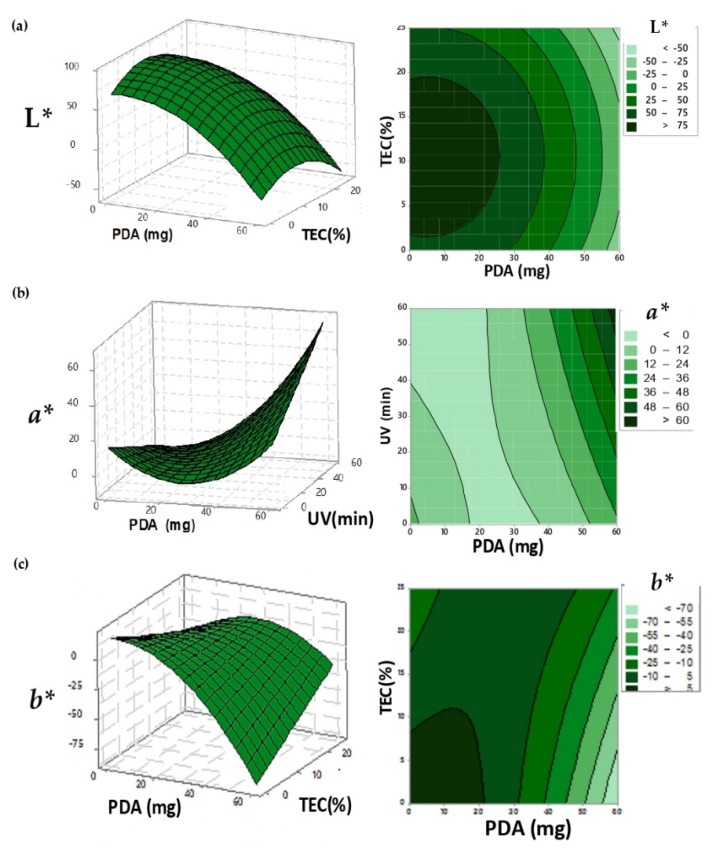
Response surface and contour graph for the surface adjusted for, respectively; (**a**) the L* value; (**b**) the *a** coordinate and (**c**) the *b** coordinate in the UV-irradiated-PDA films.

**Figure 2 biosensors-10-00059-f002:**
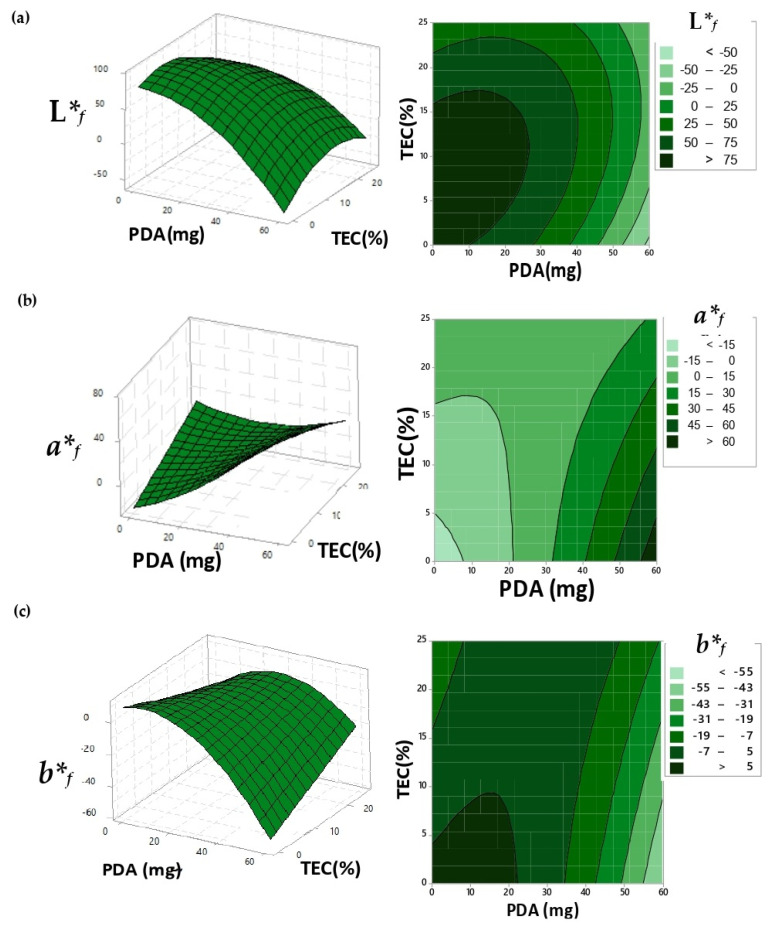
Response surface and contour graphic to the adjusted surface of the coordinates: (**a**) L**_f_*; (**b**) *a*_f_* and (**c**) *b*_f_* for the films after the exposure of UV light and different values of pH and temperature.

**Figure 3 biosensors-10-00059-f003:**
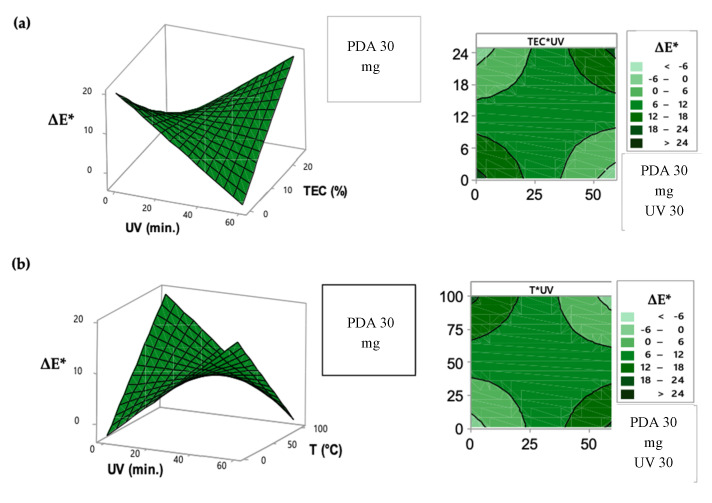
Response surface and contour graphic to the adjusted surface for the ΔE* factor as a function of: (**a**) UV-time exposure and TEC concentration; (**b**) UV-time exposure and temperature conditions on the intelligent films.

**Figure 4 biosensors-10-00059-f004:**
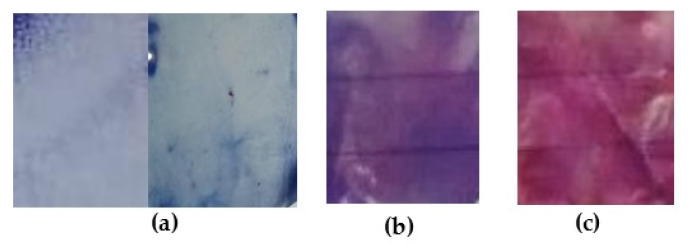
Films of cellulose acetate-based incorporated with polydiacetylene (42.63 mg) and triethyl citrate (17.73%) at pH 4.0 after 18 min of UV-light exposure: (**a**) at room temperature; (**b**) at 28.0 °C and (**c**) at 71.1 °C.

**Figure 5 biosensors-10-00059-f005:**
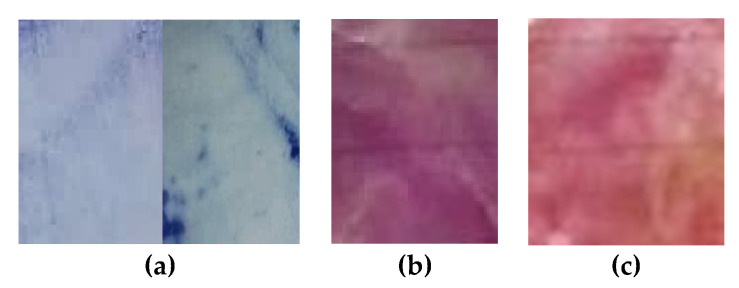
Films of cellulose acetate-based incorporated with polydiacetylene (42.63 mg) and triethyl citrate (17.73%) at 71 °C after 18 min of UV-light exposure: (**a**) at air condition; (**b**) immersed in a pH-4.0 solution and (**c**) in a pH-9.0 solution.

**Figure 6 biosensors-10-00059-f006:**
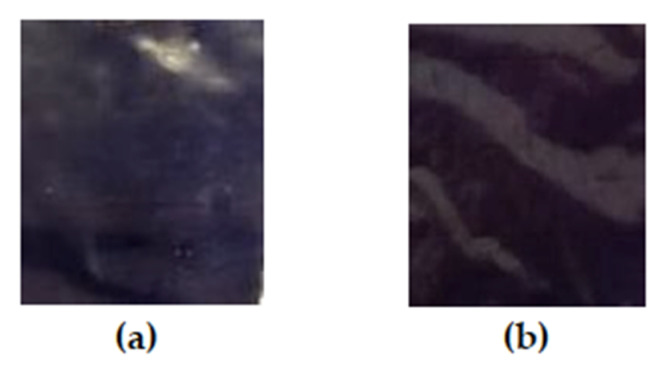
Films based on cellulose acetate incorporated with PDA (60 mg) and TEC (12.5%): (**a**) 30 min of UV-light exposure; (**b**) at pH 7.0 and 50°C condition.

**Figure 7 biosensors-10-00059-f007:**
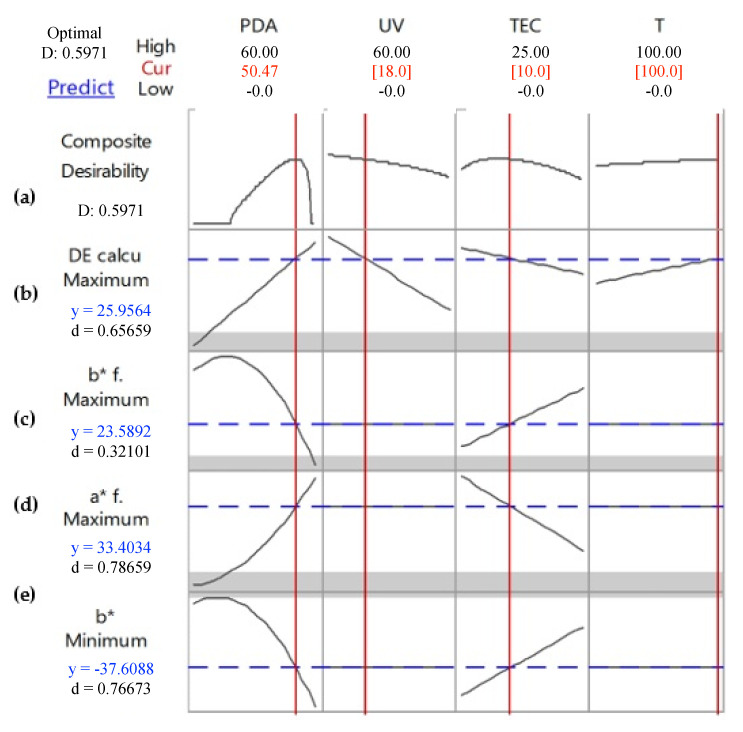
The optimized response of the intelligent films to maximize color transitioning, establishing the highest ΔE*, *a***_f_* and *b*_f_* values and the lowest *b** value of the PDA films.

**Table 1 biosensors-10-00059-t001:** Experimental range and levels of the independent variables.

Experimental Range and Levels of the Independent Variables						
Variables	Symbol Codes	Range and Levels				
		−2	−1	0	+1	2
**PDA (mg)**	X1	0.00	17.39	30.00	42.61	60.00
**TRC (%)**	X2	0.00	7.22	12.50	17.78	25.00
**Incubation Temperature (°C)**	X3	0.00	28.00	50.00	71.00	100.00
**UV (min)**	X4	1.00	18.00	30.00	43.00	60.00
**pH**	X5	1.00	4.00	7.00	9.00	11.00

**Table 2 biosensors-10-00059-t002:** Adjusted statistical models of the colorimetric behavior of polydiacetylene (PDA) films as a function of the coordinates L*, *a**, *b** and the calculated factor ΔE*.

	Equation	R^2^
**L***	68.40 + 0.36 X1 + 3.97 X2 − 0.04 X1^2^ − 0.19 X2^2^	84.67
***a****	14.90 − 1.27 X1 − 0.38 X4 + 0.02 X1^2^ + 0.02 X1 X4	87.94
***b****	18.16 + 0.01 X1 − 1.61 X2 − 0.03 X1^2^ + 0.06 X1 X2	84.81
**L**_f_***	79.30 − 0.05 X1 + 2.71 X2 − 0.04 X1^2^ − 0.19 X2^2^ + 0.05 X1 X2	82.99
***a*_f_***	U−21.34 + 0.73 X1 + 1.31 X2 + 0.01 X1^2^ − 0.06 X1 X2	74.68
***b*_f_***	9.11 + 0.34 X1 − 1.01 X2 − 0.02 X^2^ + 0.04 X1 X2	81.66
**E***	8.42 + 7.59 X1 + 2.01 X4 X2 − 1.91 X4 X3	66.14

Significative by Student’s *t*-test (*p* < 0.05).

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
