# Peer review of "Development and Evaluation of the Chromatic Behavior of an Intelligent Packaging Material Based on Cellulose Acetate Incorporated with Polydiacetylene for an Efficient Packaging"

_biosensors, 2020, doi:10.3390/bios10060059_

Round 1
Reviewer 1 Report
I have a number of concerns about this manuscript. Some of these are easily address, while others much less so.
- The manuscript requires considerable editing to improve the English. The problems go beyond simple grammatical issues to the point where some sentences become very difficult to understand,
- In line 31, the authors suggest that these films were not responsive to changes in pH, but figures 4 and 6 do seem to show a pH response both at high and low temperatures and line 268 in the conclusion also talks about the observation of pH-related changes.
- In line 89, the authors discuss the TEC concentration, but the number of significant figures in the measurement range from 2 to 4. Why is this? The same is true for the mass of PDA used.
- In line 91, the authors use the acronym DCCR without defining it. The acronym is defined in line 125 as “Rotational Central Compound Design”, but there are no references to this methodology and one would think that the acronym would be RCCD in any case. I think this is a form of Central Composite Design (CCD), but this needs to be clarified.
- The authors polymerize the PDA for various lengths of time and find, consistent with previous work, that longer irradiation time increases the extent of polymerization, but also effects the blue to red transition at longer times. However, we do not know what the light source is, nor do we have any indication of wavelength or flux.
- In line 132, the authors mention the “Response Surface Analysis”, again without providing a reference. Given the acronym “RSM”, I’m assuming they mean “Response Surface Methodology” here.
- Table 2 talks about the thickness of the films, but is not clear what the table measures. The data consist of 9 “treatments”, but there is no explanation of what these are. Are they nine randomly chosen films from table 1? Are they 9 unique films prepared with the same composition? Also, is each “treatment” the average of the 10 measurements on random spots on the same film and, if so, what is the deviation in these measurements. I can’t tell from table 2 if there are systematic changes in the film thickness based on composition or not. These questions make the claim that the films are “homogeneous” somewhat suspect.
- Line 162 should be translated to English.
- This may be a minor point, but line 183 talks about “enough energy to break the planar carbon chains of PDA”. This implies carbon-carbon bond cleavage, which is not the case. The actual process is to twist the conjugated backbone out of planarity. Related statements are made in line 203 and 221, though these are phrased better.
- In figure 4, it is not clear to me why 4(a) consists of two images, same in 6(a).
- I don’t understand what the paragraph starting on line 201 means.
- The sentence starting in line 23 talks about “steric impediment hampered Coulombic repulsion of PDA carboxylic groups” as the reason that the color change in this system is not as pronounced as with PDA liposomes. This may well be true, but it may also be that the PDA headgroups embedded in the films and inaccessible to the base, preventing deprotonation. By the way, reference 40 cited here is almost certainly incorrect. I don’t think that has anything to do with PDA polymers. Probably the authors are looking at reference 41, though the base-sensitivity of PDA vesicles has been widely reported.
- Overall, the story of this paper seems to get lost in the statistics without being terribly informative. The largest contributors to the color change are the amount of PDA, the extent of polymerization and the temperature. The first two are obvious, since the PDA-polymer backbone is the origin of the color in the first place. The sensitivity of PDA-polymers to temperature is also well known. The subtleties of TEC and pH are largely (though not entirely) lost in this analysis, since the other parameters play an outsized role. If one were interested in a temperature sensor, then the relationship between film viscoelastic properties and color change would be of interest. The rate of color changes at a particular temperature would also be a crucial feature. If a pH sensor were of interest, presumably one would want to monitor color changes at room temperature (or perhaps below) when changing the pH. This would allow an exploration of ways to optimize pH change and to understand the reduced colorimetric sensitivity in a packaging matrix without the overwhelming influence of the temperature parameter.
Author Response
Comments and Suggestions for Authors
I have a number of concerns about this manuscript. Some of these are easily address, while others much less so.
- The manuscript requires considerable editing to improve the English. The problems go beyond simple grammatical issues to the point where some sentences become very difficult to understand,
R: All English was reviewed by expertise and it is highlighted with blue color.
- In line 31, the authors suggest that these films were not responsive to changes in pH, but figures 4 and 6 do seem to show a pH response both at high and low temperatures and line 268 in the conclusion also talks about the observation of pH-related changes.
R: The pH influence on the PDA’s film color was confused and this part of abstract was changed for a better understood, such as follow: “The pH changes did not influence statistically on the color coordinates measured at analyzed ratio, but the influence of pH on the visual color changes, associated with the other factors, were observed and described”.
- In line 89, the authors discuss the TEC concentration, but the number of significant figures in the measurement range from 2 to 4. Why is this? The same is true for the mass of PDA used.
R: The TEC concentration is consolidated in polymer field and its use into cellulose acetate vary from 0 to 25%, but the amounts of PDA still has not been used into cellulose acetate and the used values was based on PDA vesicle amounts and the higher quantities tested. In the figures the range vary from 0 to 20% and the text was modified for a better understood such as follow:
“ In this study, TEC plasticizer concentration varied from 0.00% to 25.00% wt., which was pre-determined according to the literature [25]; and PDA quantities varied from 0.00 mg to 60 mg, pre-established from pre-tests done, being added into the polymeric solution in concentrations defined by Central Composite Design (CCD), as described on item 2.5.
- In line 91, the authors use the acronym DCCR without defining it. The acronym is defined in line 125 as “Rotational Central Compound Design”, but there are no references to this methodology and one would think that the acronym would be RCCD in any case. I think this is a form of Central Composite Design (CCD), but this needs to be clarified.
R: This is unclear on the text, so the text was changed for: “The compounds were added into the polymer solution according to the Central Composite Design (CCD) described on item 2.5.”
- The authors polymerize the PDA for various lengths of time and find, consistent with previous work, that longer irradiation time increases the extent of polymerization, but also effects the blue to red transition at longer times. However, we do not know what the light source is, nor do we have any indication of wavelength or flux.
R: This information was added on the text: “According to the CCD combination, the samples were irradiated with Ultraviolet light (UV) at 254 nm, constant flow, during (0, 30, 43, and 60) min, and the color measurements were done at this point.”
- In line 132, the authors mention the “Response Surface Analysis”, again without providing a reference. Given the acronym “RSM”, I’m assuming they mean “Response Surface Methodology” here.
R: You assumed well and the expression was changed to Response Surface Methodology (RSM). All the item 2.5 was rewritten for a better understood as follow:
“Central Composite Design (CCD) using Response Surface Methodology (RSM) in order to indicate the concomitant conditions for all of the factors on the colorimetric behavior analysis of PDA films. The five factors and two levels consisting of forty five experimental runs were employed including three replicates in the center point. The effects of unexplained variability in the observed response due to experimental errors were minimized by randomizing the order of the experiments, while response variables were colorimetric coordinates L*, a*, b* and the colorimetric difference determined by ΔE*. The design variables were: temperature exposure, pH values, PDA irradiation-time with UV light, PDA concentration and TEC concentration. The effects of the factors on the smart films were evaluated through RSM using Minitab 17© software. Symbol codes, experimental range and levels of the independent variables were described in Table 1..”
And Table 1 was changed for:
|
Experimental range and levels of the independent variables |
|
|||||
|
Variables |
Symbol coded |
Range and levels |
||||
|
-2 |
-1 |
0 |
+1 |
2 |
||
|
PDA (mg) |
X1 |
0.00 |
17.39 |
30.00 |
42.61 |
60.00 |
|
TRC (%) |
X2 |
0.00 |
7.22 |
12.50 |
17.78 |
25.00 |
|
Incubation Temperature (ºC) |
X3 |
0.00 |
28.00 |
50.00 |
71.00 |
100.00 |
|
UV (min) |
X4 |
1.00 |
18.00 |
30.00 |
43.00 |
60.00 |
|
pH |
X5 |
1.00 |
4.00 |
7.00 |
9.00 |
11.00 |
- Table 2 talks about the thickness of the films, but is not clear what the table measures. The data consist of 9 “treatments”, but there is no explanation of what these are. Are they nine randomly chosen films from table 1? Are they 9 unique films prepared with the same composition? Also, is each “treatment” the average of the 10 measurements on random spots on the same film and, if so, what is the deviation in these measurements. I can’t tell from table 2 if there are systematic changes in the film thickness based on composition or not. These questions make the claim that the films are “homogeneous” somewhat suspect.
R: This table does not make sense and it was removed. The film thickness was considered homogeneous because did not differ statistically in 5% of significance as described in the text: “The thickness of the intelligent films was homogeneous, and the average was 0.026 ± 0.01 µm, regardless of PDA and TEC concentration that was added into the films (p > 0.05).”
- Line 162 should be translated to English.
R: The sentence was translated: “1.Significative by Student’s T-Test (P<0,05)”
- This may be a minor point, but line 183 talks about “enough energy to break the planar carbon chains of PDA”. This implies carbon-carbon bond cleavage, which is not the case. The actual process is to twist the conjugated backbone out of planarity. Related statements are made in line 203 and 221, though these are phrased better.
R: In line 183 we have problems in english translation. Slight changes on text modified the meaningful as follow: “High temperatures increased the vibration motions that provide enough energy to provoke structural changes on PDA carbon chains”.
- In figure 4, it is not clear to me why 4(a) consists of two images, same in 6(a).
R: The figures are different and the comparison are different too. Figure 4 pictures PDA’s films after 18 min of UV-light exposure at room behavior, at pH 4.0 solution, at different temperatures 28.0 °C and a and 71.1 °C., So in Figure 4 the pH are the same but the temperature changed. Figure 6 pictures PDA’s films after 18 min exposure of UV-light at 71 ºC, and different pHs. Curiously the chromatic changes are similarly.
The figure captions were changed for a better understood:
Figure 4: “Films of cellulose acetate-based incorporated with polydiacetylene (42.63 mg) and triethyl citrate (17.73%) at pH 4.0 after 18 min of UV-light exposure a) at room behavior, b) at 28.0 °C and c) and at 71.1 °C.”
“Figure 6. Films of cellulose acetate-based incorporated with polydiacetylene (42.63 mg) and triethyl citrate (17.73%) at 71 ºC after 18 min of UV-light exposure a) at air condition, b) immersed pH 4.0 solution, and c) at pH 9.0 solution.”
- I don’t understand what the paragraph starting on line 201 means.
R: The paragraph was changed for a better understood: “Intelligent films incorporated with highest PDA concentration hampered the color transition, since the total polymerization difficult structural changes on polydiacetylene due to external stimulus such as temperature, UV or pH, therefore, requiring more energy to provoke structural change on PDA chain (Figure 5).”
- The sentence starting in line 23 talks about “steric impediment hampered Coulombic repulsion of PDA carboxylic groups” as the reason that the color change in this system is not as pronounced as with PDA liposomes. This may well be true, but it may also be that the PDA headgroups embedded in the films and inaccessible to the base, preventing deprotonation. By the way, reference 40 cited here is almost certainly incorrect. I don’t think that has anything to do with PDA polymers. Probably the authors are looking at reference 41, though the base-sensitivity of PDA vesicles has been widely reported.
R: To be right, from now, both works were cited because this affirmation was done by both. The impediment steric means this inaccessibility of PDA headgroups, so for a better understood the sentence was changed as follow: “Chen et al. (2012) [41] observed that the color transitions of vesicles produced only with tricosadinoic acid occurred after NaOH addition. The color changes occurred due to the deprotonation of the carboxylic acids, from the polymer polar groups, which increased Coulombic repulsive force, leading a new zig-zag PDA polymeric carbonic chain [42-44]. Same behavior was observed for PDA films exposed at high pH conditions, however, steric impediment due to polymer matrix hampered the deprotonation of PDA carboxylic groups reducing the intensity of the color changes.”
- Overall, the story of this paper seems to get lost in the statistics without being terribly informative. The largest contributors to the color change are the amount of PDA, the extent of polymerization and the temperature. The first two are obvious, since the PDA-polymer backbone is the origin of the color in the first place. The sensitivity of PDA-polymers to temperature is also well known. The subtleties of TEC and pH are largely (though not entirely) lost in this analysis, since the other parameters play an outsized role. If one were interested in a temperature sensor, then the relationship between film viscoelastic properties and color change would be of interest. The rate of color changes at a particular temperature would also be a crucial feature. If a pH sensor were of interest, presumably one would want to monitor color changes at room temperature (or perhaps below) when changing the pH. This would allow an exploration of ways to optimize pH change and to understand the reduced colorimetric sensitivity in a packaging matrix without the overwhelming influence of the temperature parameter.
R: This study helps to understand how the colorimetric transition caused by temperature and pH factors, well known in literature, will occurred into cellulose acetate matrix, that is unknow in science field. Besides that, slight differences such as PDA concentration, UV time of polymerization and TEC addition are important to maximize the colorimetric transition on the film to use as sensor. This study showed the amounts optimized and from this study the sensor can be improved.

Reviewer 2 Report
Recommendation: Publish after minor revisions noted
Comments:
This manuscript evaluated several factors that influence the capacity of polydiacetylene to color change into polymer matrix and used for intelligent food packaging. The title and abstract appropriately reflect the results reported. This manuscript is interesting and results should be published after minor revisions noted.
- Please address with supporting data that the colorimetric response will only occur due to the interaction between the ligand and the target and not from interferences with food compounds.
- Provide a diagram to illustrate the technique used to transfer PDA Langmuir films.
- Consider to provide an isothermal titration microcalorimetry data (time vs. molar ratio) and show enthalpy changes.
Author Response
This manuscript evaluated several factors that influence the capacity of polydiacetylene to color change into polymer matrix and used for intelligent food packaging. The title and abstract appropriately reflect the results reported. This manuscript is interesting and results should be published after minor revisions noted.
- Please address with supporting data that the colorimetric response will only occur due to the interaction between the ligand and the target and not from interferences with food compounds.
R: These studies was tested only under temperature and pH exposure. More studies will be recommended to test this interferences.
- Provide a diagram to illustrate the technique used to transfer PDA Langmuir films.
R: This study used other technique than Langmuir methodology. We added directly into a filmogenic solution the polydiacetylene compound and the film was produced using casting method. This film formed is a polymeric film and it already ready to be used as sensor. This item was unclear and for a better understood we rewrite as follow: “In this study, TEC plasticizer concentration, which varied from 0.00% to 25.00% wt., pre-determined according to values founded in literature [25], and PDA amounts, which varied from 0.00 mg to 60 mg, pre-stablished according to pre-tests done, were added into the polymeric solution in concentrations defined by Central Composite Design (CCD) was described on item 2.5.”
- Consider to provide an isothermal titration microcalorimetry data (time vs. molar ratio) and show enthalpy changes.
R: This information would be interesting, but we do not have conditions to do on time because Brazil’s academic laboratory are closed due to the pandemic situation.

Reviewer 3 Report
In this manuscript, authors have described the food packaging materials based on cellulose acetate incorporated with polydiacetylene. As the global demand for food products with natural characteristics has been increased, therefore, food packaging plays an essential role in preserving the food product quality for long-term storage for distribution and commercialization.
Thus this study is useful for food and packaging industries to improve the packaging. The work in this manuscript is well designed and therefore, may be considered for publication in the Biosensors but there are certain points that should be addressed for improving the quality of the work and the manuscript.
Since there are several grammatical errors throughout the text which downgrade the quality of the manuscript, therefore, I suggest a thorough language editing for better reading.
Additionally, the authors should examine the following points:
- 1. Line 27-29 “Film’s thickness and chromatic properties (CIELabscale) were measured, and the colour difference between films before and after exposition at pH and temperature exposition was registered for optimizing film’s composition for the best colour changes”. Authors should rewrite this sentence.
- 2. In the present manuscript, authors have studied the effect of following temperatures: 100, 50, 28 and 0°C. As several food products are also stored at -20 °C for long time preservation therefore, I suggest studying the effect of temperature below 0 °C as well (e.g., -20°C ).
- 3. In the result section authors have mentioned the thickness of films for treatment T1-T9, what about the other treatments, as each treatment has a different concentration of PDA and triethyl citrate.
- 4. Line 211; data should be corrected as ”data”.
- 5. The title of the manuscript does not read very well. The suggested title would be: “Development and evaluation of chromatic behaviour of intelligent packaging based on cellulose acetate incorporated with polydiacetylene for an efficient packaging”.
The manuscript can be accepted after the authors have carried out the suggested corrections.
Author Response
In this manuscript, authors have described the food packaging materials based on cellulose acetate incorporated with polydiacetylene. As the global demand for food products with natural characteristics has been increased, therefore, food packaging plays an essential role in preserving the food product quality for long-term storage for distribution and commercialization.
Thus this study is useful for food and packaging industries to improve the packaging. The work in this manuscript is well designed and therefore, may be considered for publication in the Biosensors but there are certain points that should be addressed for improving the quality of the work and the manuscript.
Since there are several grammatical errors throughout the text which downgrade the quality of the manuscript, therefore, I suggest a thorough language editing for better reading.
R: All English was reviewed by expertise and it is highlighted in blue.
Additionally, the authors should examine the following points:
- Line 27-29 “Film’s thickness and chromatic properties (CIELabscale) were measured, and the colour difference between films before and after exposition at pH and temperature exposition was registered for optimizing film’s composition for the best colour changes”. Authors should rewrite this sentence.
R: In this study, film thickness and film color coordinates were measured in order to study the homogeneity and the color transitioning of PDA films under different pH and temperature conditions, with the purpose of maximizing the color changes through the optimization of PDA and TEC concentrations into the cellulose acetate matrix and the polymerization degree trigged by UV light irradiation.
- In the present manuscript, authors have studied the effect of following temperatures: 100, 50, 28 and 0°C. As several food products are also stored at -20 °C for long time preservation therefore, I suggest studying the effect of temperature below 0 °C as well (e.g., -20°C ).
R: This Temperature range was chosen because low temperatures did no provoke PDA’s structural transition, so the color maintain the same, blue.
- In the result section authors have mentioned the thickness of films for treatment T1-T9, what about the other treatments, as each treatment has a different concentration of PDA and triethyl citrate.
R: This table does not make sense and it was removed. The film thickness was considered homogeneous because did not differ statistically in 5% of significance as described in the text: “The thickness of the intelligent films was homogeneous, and the average was (0.026 ± 0.01) µm, regardless of PDA and TEC concentration added into the film (p > 0,05).”
- Line 211; data should be corrected as ”data”.
R: Done.
- The title of the manuscript does not read very well. The suggested title would be: “Development and evaluation of chromatic behaviour of intelligent packaging based on cellulose acetate incorporated with polydiacetylene for an efficient packaging”.
R: Sugested accepted. Thank you.
The manuscript can be accepted after the authors have carried out the suggested corrections.

Round 2
Reviewer 1 Report
I am satisfied with the authors revisions and thank them for the obvious effort that they put in. One point was not properly addressed however. I asked about the irradiation source and the authors simply said that it was a UV lamp. That I understood, but it is necessary to know more about the light source. There are many UV lamps and they vary enormously in terms of their photon flux. Since polymerization time is one of the factors that are being measured, it is necessary to know exactly what lamp they are using.
Author Response
I am satisfied with the authors revisions and thank them for the obvious effort that they put in. One point was not properly addressed however. I asked about the irradiation source and the authors simply said that it was a UV lamp. That I understood, but it is necessary to know more about the light source. There are many UV lamps and they vary enormously in terms of their photon flux. Since polymerization time is one of the factors that are being measured, it is necessary to know exactly what lamp they are using.
R: Dear reviewer I am not sure if I could understand now but more informations about the light source were added for a better understood such as follow: “Ultraviolet light (UV) at 254 nm (constant flow, 15 W potency, 110 V, 08 x 07 x 25 cm of dimensions and 15 cm of distance work)”….
Thank you to improve the work!!! Your comments were helpful.
